# Roadmap towards Superhuman Speech Understanding using Large Language Models

## Abstract

The success of large language models (LLMs) has prompted efforts to integrate speech and audio data, aiming to create general foundation models capable of processing both textual and non-textual inputs. Recent advances, such as GPT-4o, highlight the potential for end-to-end speech LLMs, which preserves non-semantic information and world knowledge for deeper speech understanding. To guide the development of speech LLMs, we propose a five-level roadmap, ranging from basic automatic speech recognition (ASR) to advanced superhuman models capable of integrating non-semantic information with abstract acoustic knowledge for complex tasks. Moreover, we design a benchmark, **SAGI Bechmark**, that standardizes critical aspects across various tasks in these five levels, uncovering challenges in using abstract acoustic knowledge and completeness of capability. Our findings reveal gaps in handling paralinguistic cues and abstract acoustic knowledge, and we offer future directions. This paper outlines a roadmap for advancing speech LLMs, introduces a benchmark for evaluation, and provides key insights into their current limitations and potential.

## 1 Introduction

Paradigms to process *language* have been reshaped due to large language model (LLM) and its scaling law. Given the success of LLMs, one may expect to integrate extensive data in *speech* and *audio* modality into LLMs (similar to visual language models Liu et al. (2023); Li et al. (2023) [1]), resulting in a more general foundation model. Towards this path, the exploration on speech foundation models recently brings new research insights from the perspectives of multi-task and multi-lingual processing (Radford et al., 2023; Bapna et al., 2021; Zhang et al., 2023c; Seamless Communication et al., 2023; Pratap et al., 2024). A remarkable event is the release of GPT-4o, which is notable for its ability in open-ended speech-to-speech dialogue. Its performance in speech understanding, speech synthesis, and system latency has reached new levels, leading to a wave of studies on speech LLMs. The next question is, *where are we now and where should we go?* To answer this, we begin by introducing the potential of using LLMs to understand speech.

**Processing Speech using LLMs** Compared to the traditional approach of feeding ASR-transcribed text into text-only language models, unified speech-language models process raw audio or speech directly in an end-to-end fashion. The *benefits* for using LLMs to process speech are mainly two-fold. **I) Preservation of non-semantic information**: Processing raw speech directly through language models allows for the preservation of non-semantic information, such as emphasis, speaker identity, background sounds, emotions, and feelings, to the greatest extent possible. **II) World knowledge inherited in LLMs**: LLMs have superior language understanding capabilities compared to traditional models and store vast amounts of world knowledge. Therefore, starting with an LLM as the foundation for speech processing allows for the natural inheritance of this embedded knowledge, which might benefit at speech recognition level.

**Five-level Speech Understanding** The two benefits highlight the potential of speech LLMs, achieving of which requires the models to perceive complete speech information and achieve abstraction

---

[1]There exists lighweight solutions for adapting language models to process data beyond text (e.g., visual or auditory), such as: 1) using a lightweight encoder and alignment process, and 2) discretizing data into tokens, which supports the autoregressive objectives of LLMs.

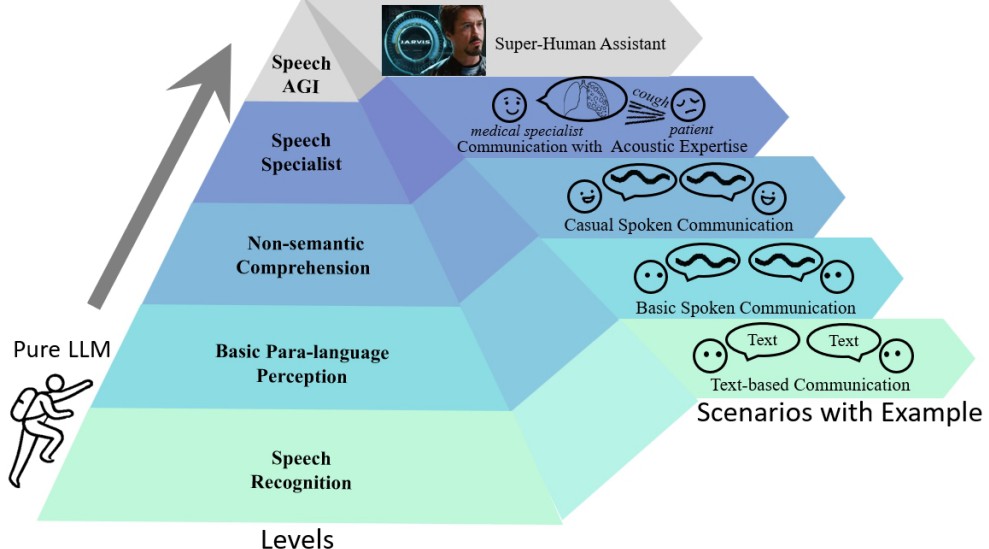

Figure 1: Levels of speech understanding using LLMs.

of expert speech/acoustic knowledge (e.g., inferring from cough and melody in some applications). To this regards, we define five levels (see Fig. 1.) as below:

- **Basic Level** At the most basic level (**Level 1**), speech language models should be able to recognize speech as text. The rationale for defining automatic speech recognition as the foundational level is that it serves as the basis for directly interacting with LLMs through speech. However, these capabilities at the basic level (e.g., speech recognition) offer limited additional benefits for ASR-equipped cascade model to understand human speech as it is somehow equivalent to a combination with a ASR model and a text-only LLM.

- **Acoustic Information Perception Levels** More advanced models (at **Level 2** and **Level 3**) are expected to directly perceive basic paralinguistic information such as tone, pitch, and loudness, and further enable them to comprehend non-semantic cues like emotions and the surrounding environment (e.g., sarcasm).

- **Abstract Acoustic Knowledge Levels** At a higher level (at **Level 4**), models can integrate speech with expert speech/audio knowledge to perform specialized tasks, such as medical assessments. At the final lavel (**Level 5**), the ultimate goal is to develop the **Speech Artificial General Intelligence (SAGI)** capable of combining non-semantic information with speech/audio knowledge to perform all speech understanding tasks, even achieving superhuman speech understanding.

**The Benchmark** However, these levels remain insufficiently intuitive. Therefore, we have preliminarily developed a benchmark to concretize and exemplify these capability levels. We designed the **SAGI Benchmark** to evaluate speech LLMs across various tasks that typically represent the characteristics of each level. The benchmark covers a wide range of tasks, including speech recognition, language distinction, volume perception, emotion recognition, and more, with each task corresponds to a specific level of capability within speech LLMs. The reliability of these evaluation sets was verified using human test, open-source and custom-trained models, demonstrating that the tasks are feasible and can be accomplished. The benchmark aims to comprehensive, tiered evaluate speech LLMs' capabilities, and exploration of their ability to apply abstract acoustic knowledge.

**Findings Human** was generally strong in tasks from level 1 to 3. However, at higher levels, human performance was limited due to a lack of abstract acoustic knowledge, which speech LLMs may start to outperform in certain tasks. **The current speech LLMs**, though capable of surpassing human performance in a few areas, still fall short in terms of task diversity and comprehensiveness. Most models struggle with even basic paralinguistic information processing, highlighting the need for further improvement. **We analyzed four reasons for performance deficiency** : 1) limited

types of training data, 2) inability to comprehensively perceive acoustic information, 3) inadequate instruction following, and 4) weak LLM backbones.

The **contributions** of this paper are as follows: We propose a *roadmap* to surpass human-level speech understanding, outlining five distinct levels to better characterize the current state of speech language models. Additionally, we design a *benchmark* aligned with this roadmap, supplementing existing benchmarks with a variety of tasks. Finally, we present key *findings* from the benchmark, based on evaluations of both speech LLMs and humans, and conduct a comprehensive *analysis* of the factors behind their suboptimal performance, offering insights and guidance for future model and architecture development.

## 2 ROADMAP TOWARDS UNDERSTANDING SPEECH

To design a roadmap for future speech LLMs, we first analyzed the development process of speech LLMs in the past (in Sec. 2.1). Following that, we present our philosophy of the roadmap in Sec. 2.2.

### 2.1 THE BACKGROUND

Current speech LLMs are mainly divided into two types: the Cascade Paradigm and the End-to-End Paradigm. Below, we will focus on analyzing these two approaches.

**Cascade Paradigm**  A straightforward approach to understanding speech using LLMs is to feed speech transcriptions (in text format) into LLMs. This is known as the *cascade* paradigm (see the left in Fig. 2). While this method allows for basic speech understanding, it lacks the ability to perceive non-semantic information (e.g., emotion, stress) within LLMs. This hinders a deeper understanding of the spoken content as its non-semantic information is often crucial for fully grasping the intent or nuances in speech.

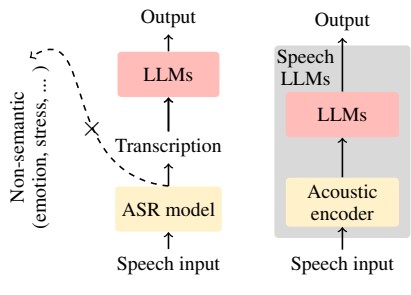

Figure 2: *Cascade* and *End-to-end* paradigms.

**End-to-end Paradigm**  In contrast, an *end-to-end* speech LLM can process both semantic and non-semantic information simultaneously within a single model. This approach not only retains more detailed information within the LLM but also allows the world knowledge embedded in the LLM to interact directly with speech data. Note that this end-to-end speech paradigm introduces additional complexity, as it requires LLMs to handle raw speech data, which operates at a lower level compared to textual inputs.

In summary, the end-to-end solution enables LLMs to directly handle non-semantic information, such as emotions. Additionally, due to its stronger perceptual capabilities, it holds greater potential for understanding and applying abstract acoustic knowledge. As a result, end-to-end solution can be considered the future direction for the development of speech LLMs.

### 2.2 THE PHILOSOPHY OF THE ROADMAP

With the rise of large language models (LLMs), there is an increasing demand to understand information beyond text, particularly speech. The core idea is that speech conveys richer information than text alone, positioning ASR (Automatic Speech Recognition) as a foundational level. End-to-end speech LLMs can begin with ASR capabilities to directly leverage the capabilities of text LLMs. And then, it progressively incorporate more advanced comprehension of non-semantic features. Finally it contains the ability to retain and apply abstract acoustic knowledge. This progress can be described as evolving through the following five levels:

**Level 1.**  *Speech Recognition Level*  At the most basic level, a speech language model should be capable of recognizing text.

These tasks form the most fundamental requirements for interacting with large models using speech. However, even at Level 1, the model offers limited advantages over a traditional cascade approach

Table 1: Levels of speech understanding using LLMs

| Level | | Semantic Information | Non-Semantic Information | Abstract Acoustic Knowledge | Remark |
|---|---|---|---|---|---|
| - | **Pure LLM** | - | - | - | Without speech input. |
| L1 | **Basic ASR** | ✓ | ✗ | ✗ | Recognizing Speech as texts. |
| L2 | **Paralinguistic Perception** | ✓ | only paralinguistic | ✗ | Perceiving direct paralinguistic *information* like tone, pitch, loudness, rhythm, and speech rate. |
| L3 | **Non-semantic Comprehension** | ✓ | ✓ | ✗ | Comprehending non-semantic *information* like speaker identity, gender, age, emotional state, and environmental sounds. |
| L4 | **Speech Specialist** | ✓ | ✓ | specialist | Understanding speech with *specific* speech *knowledge*. |
| L5 | **Speech AGI (Generalist)** | ✓ | ✓ | generalist | Understanding speech with *general* speech *knowledge*. |

(e.g., feeding ASR-transcribed text into LLMs). The real benefits of speech LLMs begin to emerge at the next level, with the ability to capture non-semantic features such as paralinguistic information.

**Level 2.** *Basic Paralinguistic Perception Level* *At this level, Speech LLMs gain the ability to perceive basic paralinguistic features in speech, such as tone, pitch, volume, rhythm, and speech rate.*

These elements are essential to speech comprehension and provide distinct advantages over pure text-based models (or Speech LLMs at Level 1). While this lays the foundation for more advanced capabilities, the insights derived at this level are still relatively shallow. For deeper understanding, we must move to Level 3, where a model comprehends a broader range of non-semantic information.

**Level 3.** *Non-semantic Comprehension Level* *At this stage, the Speech LLM extends beyond basic paralinguistic features and is capable of comprehending and interpreting more complex non-semantic information, such as emotions, sarcasm, and heightened states like pride.*

For example, emotions are higher-level human experiences that involve cognitive functions, distinguishing them from basic paralinguistic information. Interestingly, even some higher animals, like pet dogs, can perceive these types of non-semantic information. To fundamentally distinguish humans from animals, we designed Level 4 by leveraging the human strengths in higher-level cognitive capabilities.

**Level 4.** *Speech Specialist Level* *At this advanced level, Speech LLMs integrate expert-level abstract acoustic knowledge to handle a few specific, complex tasks.*

This requires integrating abstract acoustic knowledge which are advanced knowledge derived from acoustic information. This goes beyond mere recognition and comprehension at Level 1 and Level 2, requiring the model to apply higher-order thinking skills (such as analysis, evaluation, and creation) based on acoustic information [2], according to Bloom's cognitive taxonomy Krathwohl (2002). Despite these abilities, the model at this level remains domain-specific, which leads to the need for a fully generalized Speech LLM, as defined by Level 5.

**Level 5.** *Speech AGI level* *The ultimate level, Speech Artificial General Intelligence (SAGI), represents a comprehensive speech model that functions as a generalist. It can integrate knowledge from various domains and perform both general and specialized tasks, potentially surpassing human experts.*

This vision of SAGI represents the culmination of speech understanding, combining domain expertise, adaptability, and the capacity to exceed human performance in speech-based tasks. SAGI's

---

[2]This capability benefits a range of tasks, such as: 1) using cough sounds to identify the type and origin of the cough, 2) pronunciation correction, 3) music appreciation, 4) stethoscope auscultation, 5) early screening for depression and Parkinson's disease, and 6) understanding animal vocalizations.

potential to outperform humans probably stems from its ability to scale learning time and superior memory retention compared to humans. Due to time constraints, humans can typically only specialize in a narrow domain, as illustrated by '*The 10,000-Hour Rule*' in Malcolm Gladwell's book Outliers. In contrast, LLMs can easily scale their learning time by leveraging larger computing resources. Furthermore, LLMs generally possess longer memory—whether explicit or implicit—than humans, enhancing their ability to retain and apply vast amounts of information.

# 3 BENCHMARKING

## 3.1 THE NEW BENCHMARK: SAGI

To implement the roadmap (Sec.2), we aim to build a comprehensive benchmark to concretes these levels. Though previous benchmarks for speech LLMs have contributed significantly, they focus mainly on the first three levels, neglecting abstract acoustic knowledge and broader SAGI applications (App.A). Additionally, current benchmarks lack the depth needed for full speech LLM development, particularly in foundational tasks like pitch and volume perception. To address these gaps, we propose a new benchmark, detailed in the following section.

Table 2: Overview of the levels and the corresponding tasks.

| Level | Task | Dataset |
|---|---|---|
| L1 | Language Identification | Europarl-ST (Iranzo-Sánchez et al., 2020) |
| | Auto-Speech Recognition | LibriSpeech (Panayotov et al., 2015) |
| | ASR for Legal Terms* | Made of CosyVoice (SpeechTeam, 2024) |
| | ASR for Medical Terms* | Made of CosyVoice (SpeechTeam, 2024) |
| | Auto-Lyrics Transcription | Jam-Lyrics (Durand et al., 2023) |
| L2 | Volume Perception | Made of LJSpeech (Ito & Johnson, 2017) |
| | Pitch Perception | Made of SpeechAccentArchive (Weinberger, 2013) |
| | Binaural Effect Perception | Our proposed method |
| L3 | Ambient Sound Detection | Noisy speech (Valentini-Botinhao et al., 2017) |
| | Acoustic Scenes Classification | Made of MS-SNSD (Reddy et al., 2019) |
| | Speaker's Age Prediction | Made of AIR-Bench (Yang et al., 2024) & SpeechAccentArchive (Weinberger, 2013) |
| | Speaker's Gender Recognition | VCTK (Yamagishi et al., 2019) |
| | Speech Emotion Recognition | Selected from RAVDESS (Livingstone & Russo, 2018) |
| | Cappella Emotion Recognition | Selected from RAVDESS (Livingstone & Russo, 2018) |
| | Emotional Intensity Perception | Made of RAVDESS (Livingstone & Russo, 2018) |
| | Emotion Translation* | Made of RAVDESS (Livingstone & Russo, 2018) and CosyVoice (SpeechTeam, 2024) |
| | Singing Detection | RAVDESS (Livingstone & Russo, 2018) |
| L4 | COVID-19 Risk Detection | Virufy COVID-19 Open Cough Dataset (Chaudhari et al., 2020) |
| | Cough Type Classification | Made of COUGHVID(Orlandic et al., 2021) |
| | Cough Origin Diagnosis | Made of COUGHVID(Orlandic et al., 2021) |
| | Cough Severity Assessment | Made of COUGHVID(Orlandic et al., 2021) |
| L5 | Spoken English Coach | Made of speechocean762 (Zhang et al., 2021) |
| | Voice Detective | Made of SpeechAccentArchive (Weinberger, 2013) |

"*" denotes that utterances are synthesized, and the credibility verification is provided in Appendix C.5.

**Philosophy of Benchmark** The SAGI Benchmark is structured to align with the five levels of speech understanding[3], and the overview of the benchmark is shown in Tab. 2. The tasks are organized into five levels: **Level 1** focuses on testing the **recognition capabilities** of speech LLMs, including ASR, lyrics transcription, and term recognition tasks. **Level 2** evaluates **foundational perception** abilities, such as pitch and volume perception for tasks like age, gender, and emotion recognition. **Level 3** assesses **non-semantic comprehension**, incorporating tasks like emotion-integrated translation, environment perception, and emotional intensity recognition. **Level 4** explores the application of **abstract acoustic knowledge**, specifically focusing on medical-related contexts. Finally, **Level 5** envisions the capabilities of **Speech AGI (SAGI)**, highlighting tasks that promote creativity and diverse thinking, such as appreciating artwork, with a strong foundation in earlier levels.

---

[3]The types of tasks for Level 4 and 5 are not yet complete in the current version; we are working on adding more diverse tasks.

Table 3: Performance of Speech LLMs on SAGI Benchmark.

| Level | Task | Human Baseline | GPT-4o Manual Test[‡] | Models | | | |
|---|---|---|---|---|---|---|---|
| | | | | MuLLaMA | GAMA | SALMONN | Qwen2-Audio |
| L1 | Language Identification | × | 93.75% | 8.48% | × | 35.17% | 96.44% |
| | Auto-Speech Recognition | 15.49* | 11.81* | × | × | 5.45* | 4.63* |
| | ASR for Legal Terms | 98.50% | 5.00% | × | × | × | 81.04% |
| | ASR for Medical Terms | 97.50% | 35.00% | × | × | × | 53.86% |
| | Auto-Lyrics Transcription | 26.88* | × | × | × | 77.12* | 32.48* |
| | - Hallucination Rate | 3.00% | × | × | × | 29.26% | 38.21% |
| L2 | Volume Perception | 100.00% | 66.25% | 50.00% | 11.98% | 53.22% | 48.96% |
| | Pitch Perception | 96.25% | × | 33.78% | 41.5% | 50.00% | 50.00% |
| | Binaural Effect Perception | 100.00% | × | × | × | 49.88% | × |
| L3 | Ambient Noise Detection | 91.88% | 50.00% | 50.00% | 60.17% | 49.88% | 50.00% |
| | Acoustic Scenes Classification | 90.28% | × | 5.07% | 12.05% | 20.74% | 27.67% |
| | Speaker's Age Prediction | 52.59% | 35.00% | 33.60% | × | 36.87% | 38.55% |
| | Speaker's Gender Recognition | 97.50% | 65.00% | 50.00% | × | 48.12% | 79.60% |
| | Speech Emotion Recognition | 50.71% | 20.00% | 9.20% | 3.68% | 10.93% | 79.51% |
| | Cappella Emotion Recognition | 62.25% | 15.00% | 12.42% | 7.08% | 14.62% | 62.38% |
| | Emotion Intensity Perception | 97.50% | 55.00% | 50.00% | 50.00% | 49.29% | 50.00% |
| | Emotion Translation[†] | 3.68 | 0.30 | × | × | 0.27 | 0.31 |
| | Singing Detection | 99.38% | 55.00% | 50.00% | 64.82% | 56.47% | 50.22% |
| L4 | COVID-19 Risk Detection | 60.63% | × | × | × | 50.00% | 14.17% |
| | Cough Type Classification | 52.50% | × | 50.16% | 44.17% | 49.17% | 43.39% |
| | Cough Origin Diagnosis | 32.19% | × | × | × | 4.01% | 25.65% |
| | Cough Severity Assessment | 45.42% | 30.00% | 30.85% | 28.50% | 38.24% | 33.86% |
| L5 | Spoken English Coach[†] | 1.39 | 0.15 | 1.29 | 0.44 | 0.48 | 0.54 |
| | Voice Detective[†] | 1.20 | × | 0.84 | 0.83 | 0.86 | 1.24 |

"×" indicates that the model fails to follow the instruction. "*" denotes that the metric is WER (Word Error Rate) and similar, where lower values are better. "†" indicates that the task is evaluated by GPT-4, with a score ranging from 1 to 4. ‡ note that we use speech instructions to test advanced speech mode of GPT-4o, so it is not fair to directly compare it with other models, details are shown in App. B. Since GPT-4o tends to reject audio related evaluations, we only record the answers after GPT-4o responds positively to the test.

## 3.2 BENCHMARKED OBJECTS

**Humans** To conduct an initial evaluation of human performance, we created evaluation subsets by randomly selecting 80 samples per label for the objective multiple-choice tasks, and 80 samples in total for the other tasks. Four students (two males and two females) with strong English proficiency completed the assessments. The results are recorded in Tab. 3. The participant information and consistency test is in App. D.1.

**Speech LLMs** There are four types of speech LLMs, see more details in Sec. 5. We selected an open-source model for each type, except for video LLMs, where the performance on audio-only tasks is not stable. For speech-related models, we chose Qwen2-Audio for its strong performance. We selected Mu-llama for the music model and GAMA for the audio model. Additionally, we tested SALMONN as a mixed audio and speech model. We also manually tested the advanced speech mode of GPT-4o. Although it demonstrates a surprisingly interactive experience, it has not yet been tested in depth. Considering that the next step is to build the ability to follow speech instruction, we used speech instruction to test GPT-4o in conversations. For more details on model replication and evaluation settings, please refer to App. D.

## 3.3 BENCHMARKING RESULTS

**Performance for Humans** As seen in Tab. 3, human performs generally well from Level 1 to 3. However, it becomes worse at higher levels due to a lack of acoustic knowledge. On the other side, speech understanding for humans are generally better than speech language models.

**Take-away 1. *Human performance:*** *Human generally performs well in speech understanding from Level 1 to 3, but fails to reach a high level due to a lack of abstract acoustic knowledge.*

**Performance for speech LLMs** As shown in Tab. 3, speech LLMs exhibit a significant weakness in Level 2 which consists of basic listening abilities of the human. These models are currently focused on directly addressing high-level tasks while neglecting basic paralinguistic information perception,

thereby the model fails to shows generalization at higher level. Furthermore, most models do not fully satisfy the requirements at any given level, highlighting a lack of consideration for both task diversity and comprehensiveness. Notably, speech LLMs have outperformed humans in tasks like Emotion Recognition, suggesting they can discern subtle nuances beyond human perception.

**Take-away 2.** *Speech LLMs Performance: Speech LLMs still struggle with non-semantic perception and comprehension from Level 1 to Level 3, despite excelling in some tasks, limiting their performance on more complex tasks at higher levels.*

**Performance for GPT-4o** We are the first to test the understanding abilities of GPT-4o based on advanced speech mode. Although GPT-4o demonstrated novel capabilities in speech-to-speech conversations, it does not perform well in some audio understanding tasks when speech instructions are applied. On the other hand, it almost refuses to respond to audio and music-related tasks. We believe this is because speech instructions are more likely to make the model vulnerable to malicious attacks compared to text instructions.

**Take-away 3.** *GPT-4o performance: Following speech instructions is very challenging, and even GPT has significant room for improvement.*

**Future Prospects** We observe that abstract acoustic knowledge presents a common bottleneck for both humans and speech LLMs in reaching higher performance levels. Given superior capabilities of LLMs in knowledge acquisition, meanwhile, the deficiencies in diversity and completeness of capabilities can be ameliorated by incorporating additional training data. we contend:

**Take-away 4.** *Speech LLMs have the potential to exceed human capabilities, yet they currently fall short in addressing the full scope of tasks and integrating abstract acoustic knowledge.*

## 4 MORE ANALYSIS ON PERFORMANCE DEFICIENCY

In this section, we discuss reasons of performance deficiency in SAGI benchmark. We first consider composition of training data (Sec. 4.1). Then we analyse the model from three perspectives: 1) perception of acoustic information (Sec. 4.2), 2) ability of instruction following (Sec. 4.3), and 3) capacity of LLM backbone (Sec. 4.4).

### 4.1 LIMITED TYPES OF TRAINING DATA

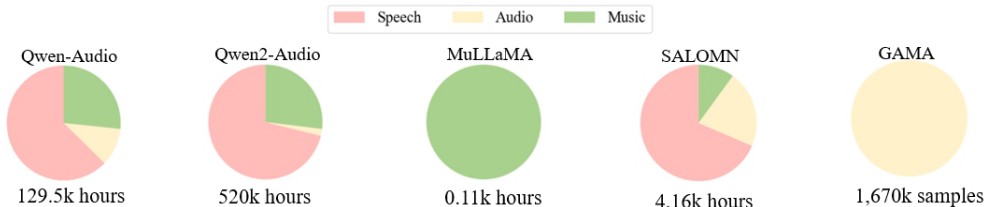

Figure 3: Components of three types of training data.

We observed in Tab. 3 that certain tasks, particularly those in Level 2, are easy for humans but challenging for speech LLMs. We first analyzed the composition of the training data for speech LLMs, as shown in Fig. 3. We found that most speech LLMs tend to disregard audio data except for GAMA, whereas GAMA focuses primarily on audio. This indicates that the data bias across different speech LLMs is distinct, which subsequently leads to variations in task preference.

To further examine the influence of task preference, we compared the performance of various speech LLMs with Whisper V3 (trained with ∼5,000k hours), as shown in Tab. 4. We find that Whisper still outperforms other models on the Lyrics Transcription task, benefiting from the massive training data. On the other hand, with the help of the learned knowledge, speech LLMs perform significantly better at recognizing certain terms. This demonstrates that speech LLMs have great potential compared to traditional speech models. Notably, we also tested a small model trained exclusively on an audio dataset. This small model achieved 100% accuracy, while speech LLMs struggled with the task.

**Take-away 5.** *Current insufficient diversity and completeness of training data could not help speech LLMs reach a higher level.*

Table 4: Comparison of speech model and LLMs. The small model uses Transformer with 10M parameters.

| Subtask | Task type | Model | Result | Best result of LLMs |
|---|---|---|---|---|
| Language Identification | 5-Categories | Whisper | 91.45% | 96.62% |
| Auto-Speech Recognition | Generation | Whisper | 2.44 | 2.65 |
| ASR for Legal Term | Generation | Whisper | 33.33% | 81.04% |
| ASR for Medical Term | Generation | Whisper | 34.98% | 53.86% |
| Auto-Lyrics Transcription | Generation | Whisper | 22.10 | 32.48 |
| Hallucination Rate | 2-Categories | Whisper | 14.63% | 29.26% |
| Volume Perception | 2-Categories | Small model | 100.00% | 53.22% |

## 4.2 INABILITY TO COMPREHENSIVELY PERCEIVE ACOUSTIC INFORMATION

The current end-to-end paradigm almost universally adopts the stacking paradigm. But the stacking paradigm may suffer from two types of information loss: 1) the latent representation produced by the acoustic encoder does not fully capture or convey the necessary information, and 2) the acoustic encoder fails to transfer all the information to the downstream LLMs.

We first investigate whether the loss of latent representation contributes to the limited performance. We compare the speech features generated from the same text content, but spoken by different genders and with different emotions. The features are generated by Whisper and analyzed using cosine similarity between the original and perturbed speech. The results, shown in Fig. 4, indicate that there is no significant difference between different speech samples. This suggests that emotion and gender information are lost during the acoustic encoder process. This could explain why some speech LLMs perform poorly on certain simple tasks, as the LLMs cannot compensate for the loss of acoustic information.

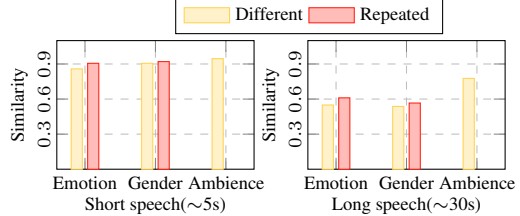

Figure 4: Representation similarity of different speeches. Each speech pair has the same content but is spoken with different style. The representation is generated by the Whisper encoder.

We then assess whether information loss from the acoustic encoder to downstream LLMs limits speech LLMs' performance. We choose the base cases of the ASR task where the WER (Word Error Rate) is higher than 20%, as shown in Tab. 5. We found that the error types is different between the whisper and speech LLMs. Considering that Qwen-Audio is built on Whisper, the results confirm that LLMs cannot correct errors from the acoustic model. A typical difference between Whisper and speech LLMs is the occurrence of overlong generation, which is a form of hallucination.

Table 5: Two types of recognizing error. The "truncation" and "over-long" denote the generation is short and longer than the length of reference more than 20% separately.

| Model | Total | Truncation | Over-long |
|---|---|---|---|
| Whisper | 64 | 3 | 0 |
| Qwen-Audio | 68 | 5 | 6 |
| Qwen2-Audio | 149 | 89 | 3 |
| SALMONN | 251 | 154 | 5 |

Another notable phenomenon is that almost 60% of error cases are caused by truncation. Additionally, we observed that the speech LLMs sometimes omits the start of a sentence, which does not happen with Whisper. This prove that speech LLMs suffer the loss of information transfer between the LLMs and the acoustic encoder. The current stacked paradigm often tunes base on LLMs with most parameters frozen, which requires the acoustic features to fit the LLMs' representation space. This requirement hinders the seamless transmission of acoustic information to the LLMs, leading to premature termination of the generation process.

**Take-away 6.** *LMs in Current End-to-end solutions fail to encode complete acoustic information.*

## 4.3 INADEQUATE INSTRUCTION FOLLOWING

We have observed that some models exhibit poor instruction following in Tab. 3. Two reasons can lead to these results: 1) the models do not understand the instructions, and 2) the instruction fails to help the models comprehend the speech.

We classify the cause by observing changes in performance after perturbing the prompt. If the model is insensitive to different perturbed prompts, it indicates that the model cannot understand the prompt. On the other hand, if the models show significantly better performance with a properly structured prompt, it suggests that the model could understand the task, while requires the specific instruction. We choose the two Level 3 tasks (Age prediction and Ambient Noise Detection) where the instruction following ability is crucial, and the results shown in Fig. 5.

For the result of Fig. 5, we can find the Mullama is not sensitive about the instruction. This prove the model can not figure out this task. Further,

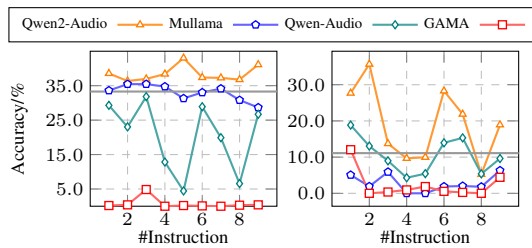

Figure 5: Performance of speech LLMs with different instruction on speaker age task (left) and scenes classification task (right).Gray line shows random selection accuracy. Details about the instructions and results are shown in App. E.

the performance of most speech LLMs highly related with the specific prompt, this shows models are sensitive with the format of instruction. Comparing with the text LLMs which are robust with diverse instruction, the speech LLMs need much effect to guarantee instruction following.

**Take-away 7.** *Current speech LLMs follow instructions poorly.*

## 4.4 WEAK LLM BACKBONES

Table 6: There different tasks to test the ability of processing the phone

| Task | Prompt |
|---|---|
| Sequence-level | Given a phone sequence, "M AA0 R K IH0 Z ...", what sentence does it represent? |
| Token-level | Given a tokenized phone sequence, "[M AA0 R K] [IH0 Z] ...", what sentence does it represent? |
| Token-level with one shot | Given a tokenized phone sequence, "[M AA0 R K] [IH0 Z] ...", what sentence does it represent? For example, if the phone sequence is "[F AO0 R] [F AY0 V], [S IH0 K S] [S EH1 V N] [EY0 T]" the sentence can be: "four five six seven eight nine". |

Most current speech LLMs follow the paradigm of stacking the acoustic model and text LLMs. This paradigm requires the text LLMs to process audio-like tokens, raising an intuitive question: whether text LLMs have the potential to handle cross-modal tasks. We designed a direct task of converting a phoneme sequence into a complete sentence. The phoneme represents pronunciation in text format, thus understanding phonemes can demonstrate the model's potential to process audio. We designed three different tasks, as shown in Tab. 6. The most challenging task requires the model to find the most likely sentence according to the entire phoneme sequence, which takes some time even for humans.

Table 7: Potential of LLMs to process speech. The metric is WER, and if LLMs show the hallucination or reject to answer, we calculate the WER with 100% for this case.

| Model | Seq. ↓ | Token ↓ zero-shot | Token ↓ one-shot |
|---|---|---|---|
| GPT-4o | 17.5 | 8.3 | 8.3 |
| Mixtra-7B | 99.5 | 98.9 | 97.7 |
| Qwen2-7B | 99.3 | 98.3 | 95.8 |
| Llama3-7B | 97.5 | 89.6 | 87.9 |
| Llama3.1-8B | 94.0 | 83.7 | 78.0 |
| Mixtra 8x7B | 98.2 | 95.1 | 92.6 |
| Qwen2-72B | 93.4 | 75.4 | 73.5 |
| Llama3.1-70B | 80.5 | 51.1 | 46.9 |

We evaluate the most commonly used LLMs for building speech LLMs, and the results are shown in Tab. 7. We found that the closed-source GPT-4o demonstrates a surprising ability to process

phonemes, proving that it can easily be converted into a powerful speech LLM. On the other hand, all the open-source models fail to show potential in handling audio. Even when the size of the model parameters is increased, the ability remains very limited.

One explanation is that open-source models overlook potential audio-related tasks, which is quite unlike GPT-4o. This leads to a significant gap between the two types of models. A piece of evidence supporting this is that Llama 3.1, which emphasizes multi-modal capabilities Dubey et al. (2024), shows a noticeable improvement in WER in token-level tasks and delivers robust performance with 70B parameters. Overall, open-source foundation models still have substantial room for improvement in their ability to handle audio-related tasks.

**Take-away 8.** *The used LLM backbone is relatively weak for current speech LLMs.*

## 5 RELATED WORK

Speech language models have seen a surge in development following the advent of LLMs. Currently, most work integrates pre-trained acoustic models with LLMs using an alignment module. There are two main strategies to bridge the gap between the two models: 1) adapters and 2) attention mechanisms.

**Adapter** The former method adds modules (usually convolutional networks and MLPs) between the acoustic model and LLMs. Convolutional networks can compress sequence length (Wang et al., 2023a), while MLPs are used to align acoustic tokens with text embeddings (Su et al., 2023).

**Attention Mechanisms** Regarding the attention method, Kong et al. (2024) implemented cross-attention to filter information from the output of the speech encoder. Li et al. (2023) proposed the Q-former as an intermediate extractor based on cross-attention. Similarly, Pan et al. (2023) applied the Q-former to extract useful acoustic information for LLMs. Some works directly treat the acoustic codec as tokens and do not rely on alignment strategies (Zhang et al., 2023a; Rubenstein et al., 2023).

**Categorization of speech LLMs** We have introduced that acoustic models can generally be divided into four types. Some works aim to build **universal multi-modal LLMs** (Su et al., 2023; Zhan et al., 2024; Wu et al., 2023b; Lyu et al., 2023; Zhang et al., 2023b; Shukor et al., 2023). Several studies focus on enhancing **music understanding**, an important area that has not yet received enough attention (Deshmukh et al., 2023; Zhan et al., 2024; Liu et al., 2024a). Most speech LLMs aim to improve **speech-to-text tasks** and **multi-turn dialogue capabilities** (Fathullah et al., 2024; Shu et al., 2023; Wang et al., 2023b; Pan et al., 2023; Rubenstein et al., 2023; Zhang et al., 2023a; Bai et al., 2024; Wu et al., 2023a; Maiti et al., 2024; Wang et al., 2023a; Chu et al., 2024; Dubey et al., 2024). Some works utilize audio codec models to enhance audio processing performance (Chen et al., 2023; Kong et al., 2024; Nguyen et al., 2024; Das et al., 2024; Gong et al., 2023). Inspired by these efforts, several studies (Tang et al., 2023; Ghosh et al., 2024a; Hu et al., 2024) combine acoustic and semantic codecs to integrate audio and speech processing capabilities into a single model.

## 6 CONCLUSION

In this paper, we explored the evolving landscape of large language models (LLMs) in the realm of speech processing. We introduced a five-level roadmap to guide the development of human-level speech understanding, from basic ASR capabilities to advanced generalist models that integrate non-semantic information with general abstract acoustic knowledge for complex tasks. To assess the current state of speech LLMs, we designed a comprehensive benchmark that standardizes critical aspects across various tasks, ensuring consistency and reliability in performance evaluation. Our research reveals the current stage and deficiencies in understanding speech by both humans and speech LLMs. We evaluate the advanced speech mode of GPT-4o and find that following speech instructions is very challenging. Further analysis has uncovered structural flaws in existing speech LLMs. Reveals that current speech LLMs face issues in both Acoustic Information Transfer and Foundation LLMs' Potentiality. The contributions of this paper provide a structured approach to advancing speech LLMs, offering valuable insights for future innovations in this field.

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

## LIMITATION

Artificial intelligence should not be confined to overly narrow domains, as such a focus can lead to frequent model switching when handling diverse tasks. This requires SAGI, a speech AGI, to be a powerful assistant capable of completing all kinds of tasks. However, during our primary testing, most speech LLMs remain at levels 1 and 2, indicating there is still a long way to go in terms of understanding speech.

To advance further, we conclude some important directions for improving speech LLMs toward higher level:

- Requiring more diverse speech data to handle complex tasks.
- Enhancing the ability of text LLMs to process speech-related tasks.
- Ensuring that LLMs can receive complete acoustic information.

We advocate for the development of more powerful acoustic models, consideration of cross-domain compatibility when constructing datasets, and a deepening of expertise in specific research areas. This approach will enhance the generalization and adaptability of the models.

## A  EXISTING BENCHMARK

Table 8 summarizes the coverage of existing benchmarks across different levels of speech model tasks, highlighting gaps in current evaluation methods. L1 tasks such as Speech ASR, Intent Classification, and Language Identification are well supported by both Dynamic-SUPERB and AIR-Bench, though SD-Eval lacks coverage. For L2 foundational perception tasks, like Music Pitch and Velocity, only AIR-Bench provides support. L3 tasks related to non-semantic comprehension, such as Emotion, Environment, and Speaker Gender/Age, are covered to varying degrees across all benchmarks, with Dynamic-SUPERB offering the most comprehensive support. However, more specialized tasks like Sarcasm, Stress, and Spoof Detection are only covered by Dynamic-SUPERB. Notably, L4 (Abstract Knowledge) and L5 (Speech AGI) remain entirely unsupported across all benchmarks. This underscores the urgent need to build a more comprehensive benchmark that addresses the gaps in L2, L4, and L5, ensuring more robust evaluation across all levels of speech model tasks.

Table 8: Existing benchmarks across Levels. L2, L4 and L5 have not received enough attention yet.

| Level | Task | Dynamic-SUPERB | AIR-Bench | SD-Eval |
|---|---|:---:|:---:|:---:|
| L1 | Speech ASR | ✓ | ✓ | ✗ |
| | Intent Classification | ✓ | ✓ | ✗ |
| | Language Identification | ✓ | ✓ | ✗ |
| L2 | Music Pitch and Velocity | ✗ | ✓ | ✗ |
| L3 | Emotion | ✓ | ✓ | ✓ |
| | Environment | ✓ | ✓ | ✓ |
| | Accent | ✓ | ✗ | ✓ |
| | Speaker Gender/Age | ✗ | ✓ | ✓ |
| | Noise Detection | ✓ | ✗ | ✗ |
| | Speaker Verification | ✓ | ✓ | ✗ |
| | Sarcasm Detection | ✓ | ✗ | ✗ |
| | Stress Detection | ✓ | ✗ | ✗ |
| | How Far Are You | ✓ | ✗ | ✗ |
| | Spoof Detection | ✓ | ✗ | ✗ |
| | Synthesized Voice Detection | ✗ | ✓ | ✗ |
| L4 | No Related Work | ✗ | ✗ | ✗ |
| L5 | No Related Work | ✗ | ✗ | ✗ |

## B  CHALLENGING SPEECH INSTRUCTION

Due to the significant discrepancy between the objective test results of GPT-4o and our intuitive impressions, we were motivated to explore whether following speech instructions is more challeng-

ing. Since GPT-4o can only be triggered through speech instructions, we conducted experiments with Qwen2-Audio, which supports both methods. The results, as shown in Tab. 9, indicate that following speech instructions is indeed more challenging.

Table 9: Comparison of text instruction and speech instruction.

|  | Language Identification ↑ | ASR ↓ | ASR for Medical Terms↑ | ASR for Legal Terms↑ |
|---|---|---|---|---|
| Text Instr. | 96.44% | 4.63 | 53.86% | 81.04% |
| Speech Instr. | 48.98% | 7.72 | 55.17% | 74.51% |

As established, speech instructions are more challenging. So, when using speech instructions, which is stronger: Qwen2-Audio or GPT-4o? To ensure a fair comparison, We tested Qwen2-Audio on the same test subset used for GPT-4o. The results are shown in Tab. 10.

Table 10: Comparison of Qwen2-Audio and GPT-4o with speech instruction.

|  | Language Identification | ASR | ASR for Medical Terms↑ | ASR for Legal Terms↑ |
|---|---|---|---|---|
| Qwen2-Audio | 47.00% | 20.02 | 65.00% | 85.00% |
| GPT-4o | 94.00% | 11.81 | 35.00% | 5.00% |

## C  DETAILS OF BENCHMARK CONSTRUCTION

The overall construction principles are provided in Sec.C.1. The data and tools used are detailed in Sec.C.2. The composition structure of the data is outlined in Sec.C.3. Detailed construction details for each task are available in Sec.C.4.The credibility verification of synthesized speech is provided in Sec. C.5.

### C.1  GENERAL PRINCIPLES OF DATA CONSTRUCTION

#### C.1.1  QUESTION CONSTRUCTION

For objective multiple-choice questions, we included multiple-choice options in the questions to guide large models in generating the final results. For subjective response questions, we specified the main aspects around which the questions revolve and set suggested answers, although these do not require the model to produce results that are exactly identical, illustrated in Fig. 6.

#### C.1.2  UNIFORM SAMPLING RATE

To ensure that these evaluation results truly reflect the differences in the model's performance across various tasks without being influenced by the audio sampling rate, and considering that increasing the sampling rate of audio data may introduce additional errors, this paper chooses to align all datasets to the one with the lowest sampling rate. Therefore, all test data is downsampled to 16,000 Hz.

#### C.1.3  UNIFORM NUMBER OF AUDIO CHANNELS

To standardize the format of the input audio, we converted all audio files for the tasks into mono channel, except for those in the Binaural Effect Perception sub-task.

#### C.1.4  UNIFORM AUDIO DURATION

Most speech LLMs employ the encoder from Radford et al. (2023), which is designed to handle a maximum duration of 30 seconds. Consequently, the processing length for the majority of speech LLMs is capped at 30 seconds. To ensure a level playing field for all speech LLMs, we have restricted the audio lengths to a maximum of 30 seconds.

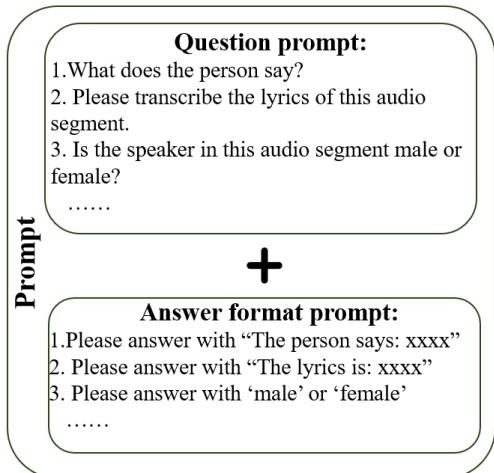

Figure 6: The method for generating the problem prompts.

### C.1.5 UNIFORM OPTION RATIO

For the binary classification problem, we performed data balancing so that both options account for 50% of the data. Due to some limitations in the current models, they might always choose one option in binary classification tasks. If the data were unbalanced, such as 40% for one option and 60% for the other, different models that always pick the same option could yield very different results, even though their capabilities are similar. This is not what we want, so we balanced the data for all binary classification tasks.Please refer to Tab. 11 for detailed information.

### C.2 DATA AND TOOLS UTILIZED

We used the following 10 datasets:

Europarl-ST (Iranzo-Sánchez et al., 2020) ,LibriSpeech (Panayotov et al., 2015),JamendoLyrics MultiLang dataset (Durand et al., 2023), LJSpeech (Ito & Johnson, 2017),Noisy speech (Valentini-Botinhao et al., 2017),SpeechAccentArchive (Weinberger, 2013) ,VCTK (Yamagishi et al., 2019),RAVDESS( (Livingstone & Russo, 2018),AISHELL-MDSC (Gao et al., 2024),spee-chocean762 (Zhang et al., 2021)

We utilized two open-source tools: MS-SNSD (Reddy et al., 2019),cosyVoice (SpeechTeam, 2024)

### C.3 DATA STRUCTURE OF BENCHMARK

Data samples are represented as (P, Q, A, D), where P denotes the audio path, Q represents the question, A corresponds to the answer, and D provides additional explanations to aid researchers in understanding the data.

### C.4 DETAILS OF EACH TASK

### C.4.1 LANGUAGE IDENTIFICATION

We used Europarl-ST (Iranzo-Sánchez et al., 2020) to construct our evaluation dataset.Europarl-ST is a multilingual speech translation corpus containing paired audio-text samples for speech translation. It was constructed using debates held in the European Parliament between 2008 and 2012. We selected five commonly used languages: German, English, French, Spanish, and Italian, with 500 data entries for each language. In order for all models to be able to read in this data and ensure fairness, we used the first 30 seconds of each audio clip as the actual input.The task was set as: "What language is spoken in this audio segment?Please choose from the German, English, French, Spanish, and Italian options?"

Table 11: Utterances for Each Task

| Task | Utterances |
|------|------------|
| Language Identification | German: 500, Spanish: 500, English: 500, French: 500, Italian: 500 |
| Auto-Speech Recognition | English:2791 |
| ASR for Legal Terms | Chinese:102 |
| ASR for Medical Terms | Chinese:203 |
| Auto-Lyrics Transcription | English: 868 |
| Volume Perception | increasing: 512, decreasing: 512 |
| Pitch Perception | (80-150)Hz: 300, (180-250)Hz: 300 |
| Binaural Effect Perception | left ear: 400, right ear: 400 |
| Ambient Noise Detection | has: 824, hasn't: 824 |
| Acoustic Scenes Classification | Babble: 310, CopyMachine: 310, Neighbor: 310, ShuttingDoor: 315, AirportAnnouncements: 305, Munching: 300, Typing: 310, AirConditioner: 305, VacuumCleaner: 310 |
| Speaker's Age | teens to twenties: 330, thirties to forties: 330, fifties to sixties: 330 |
| Speaker's Gender | female: 1410, male: 1410 |
| Speech Emotion Recognition | happy: 200, disgust: 200, fearful: 200, sad: 200, surprised: 200, angry: 200, neutral: 100 |
| Cappella Emotion Recognition | angry: 184, sad: 184, happy: 184, fearful: 184, neutral: 92 |
| Emotion Intensity Perception | former: 150, latter: 150 |
| Singing Detection | singing: 1012, speech: 1012 |
| Cough Type Classification | wet: 300 , dry: 300 |
| Cough Origin Diagnosis | COVID-19: 198, healthy cough: 200, lower infection: 200,upper infection: 200 |
| Cough Severity Assessment | pseudocough: 170, mild: 170, severe: 170 |

## C.4.2   AUTOMATIC SPEECH RECOGNITION

We constructed our evaluation dataset based on LibriSpeech (Panayotov et al., 2015).Inspired by Radford et al. (2023), we used the test-clean and test-other splits as our test sets,a total of 2791 data entries. It should be noted that the corresponding text of LibriSpeech consists of uppercase letters. Since we standardized the text during WER computation, as detailed in D.4.1, this will eliminate the impact of these uppercase letters. Therefore, we did not perform any additional processing when constructing the dataset.The task was set as:What does the person say?Please answer with "The person says: xxxx".

## C.4.3   ASR FOR LEGAL TERMS

We selected 27 offenses defined under Chinese criminal law and combined them with four templates to generate 108 sentences, which were synthesized using cosyVoice (SpeechTeam, 2024). After manual screening (detailed in Sec. C.5.1), 102 utterances remained.The task was set as:What does the person say?Please answer with "The person says: xxxx".This approach is consistent with ASR, as we believe that this ability should be demonstrated automatically during the ASR process without the need for additional prompts.

## C.4.4   ASR FOR MEDICAL TERMS

We selected 62 medical terms referring to specific locations and combined them with four templates to generate 248 sentences, which were synthesized using cosyVoice (SpeechTeam, 2024). After manual screening (detailed in Sec. C.5.1), 203 utterances remained.The task was set as:What does the person say?Please answer with "The person says: xxxx".This approach is consistent with ASR, as we believe that this ability should be demonstrated automatically during the ASR process without the need for additional prompts.

### C.4.5 AUTOMATIC LYRICS TRANSCRIPTION

We utilized the JamendoLyrics MultiLang dataset (Durand et al., 2023) for our research. We acknowledge that a revised version of this dataset has been released as the Jam-Alt dataset (Cífka et al., 2023). However, in accordance with the constraints outlined in Sec. C.1.4, we were required to resegment the audio files. Given that the Jam-Alt dataset, as described by its authors, exhibits certain deviations in its timestamps, we elected to employ the JamendoLyrics MultiLang dataset as our primary dataset for construction purposes. During the construction process, we manually selected the segmentation points and employed code to segment the audio files, thereby obtaining our final dataset.The task was set as: "Please transcribe the lyrics of this audio segment.Please answer with:The lyrics is: xxxx"

### C.4.6 VOLUME PERCEPTION

We constructed our evaluation dataset based on LJSpeech (Ito & Johnson, 2017). Following the data split of Chien et al. (2021), we used 512 test samples. We set up two scenarios: one where the volume gradually increases from 0 to its original level, and another where it decreases from the original level to 0. We tasked the model with determining whether the volume is increasing or decreasing.The task was set as: "Is the volume of this audio segment gradually increasing or decreasing?"

### C.4.7 PITCH PERCEPTION

We used the SpeechAccentArchive (Weinberger, 2013) dataset to construct our test set. During this process, we first identified the frequency ranges with the highest proportion of fundamental frequency (F0). Ultimately, we selected the ranges (80, 150) Hz and (180, 250) Hz for our experiments. We framed the problem as follows: "In the following audio segment, into which range does more than 70% of the fundamental frequency content fall? Please choose from the following two ranges: (80, 150) Hz and (180, 250) Hz." We calculated the proportion of F0 content falling within these two ranges for each audio segment and selected the corresponding data. During the process, we ranked all the data, prioritizing those segments with a higher proportion.

### C.4.8 BINAURAL EFFECT PERCEPTION

We generated random sounds using four methods: sine wave, square wave, triangle wave, and noise. These sounds are heard only in the left ear or the right ear. For more details, please refer to our public code. The model is used to determine which ear hears these sounds.The task was set as: "In this audio segment, does the sound appear in the left ear or the right ear?Please answer with 'left' or 'right'."

### C.4.9 AMBIENT NOISE DETECTION

We constructed the evaluation dataset using Noisy speech (Valentini-Botinhao et al., 2017).Noisy speech dataset contains corresponding pairs of noisy and clean data. The purpose of the dataset is to explore methods for speech enhancement.We selected the entire test set from this dataset, which includes 824 clean audio clips and 824 audio clips with ambient noise. We used all of these data, and the task was set as: "Is there any ambient noise in this audio segment, in addition to the speaker voice?Please answer with yes or no."

### C.4.10 ACOUSTIC SCENES CLASSIFICATION

We used MS-SNSD (Reddy et al., 2019) to synthesize these test datasets.MS-SNSD is a tool for synthesizing speech with environmental noise, aimed at advancing research in speech enhancement. We selected 51 environmental noise samples from its test set to synthesize 6,105 test samples, and the task was set as: "What is the ambient noise of this audio segment? Please choose from the ['Babble', 'CopyMachine', 'Neighbor', 'ShuttingDoor', 'AirportAnnouncements', 'Munching', 'Typing', 'AirConditioner', 'VacuumCleaner'] options?"

### C.4.11 SPEAKER'S AGE PREDICTION

We have observed that there are relatively few datasets specifically aimed at speaker age recognition. We noted that the AIR Bench (Yang et al., 2024) has done an excellent job in addressing this task,We followed their approach of categorizing age into four groups but noticed that their data distribution was not balanced, specifically: teens to twenties: 653, thirties to forties: 268, fifties to sixties: 64, seventies to eighties: 15. Therefore, we used the SpeechAccentArchive (Weinberger, 2013) to balance the age distribution. Unfortunately, we found it difficult to obtain sufficient data for the seventies to eighties category, so we retained only three categories: teens to twenties, thirties to forties, and fifties to sixties.And the task was set as: "Which age range do you believe best matches the speaker's voice? Please choose from the ['teens to twenties', 'thirties to forties', 'fifties to sixties'] options?"

### C.4.12 SPEAKER'S GENDER RECOGNITION

We constructed the evaluation dataset using VCTK (Yamagishi et al., 2019).To balance the number of males and females in the benchmark, considering there are 61 female speakers and 47 male speakers in the VCTK dataset, we selected the top 47 female speakers along with all the male speakers. For each speaker, we chose the first 30 audio recordings.The task was set as: "Is the speaker in this audio segment male or female?Please answer with 'male' or 'female'"

### C.4.13 SPEECH EMOTION RECOGNITION

In a genuine sense, understanding emotions in models should not solely depend on interpreting text. Emotions do not have a one-to-one correspondence with sentences; the same sentence can express various emotional tones depending on the speaker's emotional state. Therefore, it is crucial to advocate for models to move beyond mere textual content of sentences when inferring emotions and to delve into the non-textual information within the speech. Accordingly, in the evaluation set for emotion recognition, we employed a dataset unrelated to both the emotions and the sentence content—the RAVDESS dataset (Livingstone & Russo, 2018).The task is then defined as: "What emotion does this audio clip convey?Please answer by single word select from ['neutral', 'happy', 'sad', 'angry', 'fearful', 'disgust', 'surprised']."

To demonstrate that the emotions in our constructed dataset are independent of the textual content, we used a combination of the whisper-v3-large (Radford et al., 2023) model and the gpt-4-o (OpenAI, 2023) model to predict the emotions in the audio files of the dataset. The experimental results can be found in the Tab. 12

Table 12: emotion detection evaluation set Supplementary experiments

|          | **First repetition** | **Second repetition** | **Third repetition** |
|----------|----------------------|-----------------------|----------------------|
| accuracy | 10.53%               | 9.33%                 | 9.73%                |

### C.4.14 CAPPELLA EMOTION RECOGNITION

We also used RAVDESS ( (Livingstone & Russo, 2018)) to construct the evaluation set for singing emotion detection.The task is then defined as: "What emotion does this audio clip convey?Please answer by single word select from ['neutral', 'happy', 'sad', 'angry', 'fearful', 'disgust', 'surprised']."

### C.4.15 EMOTIONAL INTENSITY PERCEPTION

We used the RAVDESS ( (Livingstone & Russo, 2018)) dataset to construct the evaluation set for Emotional Intensity Perception. Since most models accept only a single audio input, we merged two audio segments and tasked the model with analyzing which part of the combined audio segment exhibits stronger emotional intensity. Specifically, we defined the problem as follows: "In this audio segment, a sentence is repeated twice. Is the emotion in the 'former' stronger or the 'latter' stronger? Please answer with 'former' or 'latter.'"To balance the proportion between the two options, we alternated the placement of the stronger emotion, sometimes positioning it at the former and other times at the latter when synthesizing the data.

### C.4.16 EMOTION TRANSLATION

We believe that translations should reflect different expressions based on the emotional context. For example, the phrase "What are you doing?" can convey various meanings depending on the emotion—whether it's anger, surprise, sadness, or neutrality. In an angry context, it expresses strong disapproval or questioning of the person's actions; in a surprised context, it conveys disbelief about what the other person is doing; and in a sad context, it should reflect disappointment. Therefore, translations should be adjusted accordingly to better capture these nuances.

We observed that cosyVoice (SpeechTeam, 2024) demonstrates excellent zero-shot capabilities, effectively mimicking the tone and style of the input speech prompt. Therefore, we used cosyVoice to emulate the sentences with strong emotions from the RAVDESS (Livingstone & Russo, 2018) dataset to generate speech with corresponding emotions. After synthesis, we had five native speakers review the generated speech. If any of the native speakers felt that the synthesized speech did not convey the intended emotion, that segment was discarded. Ultimately, we obtained xxx valid speech samples.and the task was set as: "Please translate the following sentence into the most appropriate Chinese, based on the emotion and content of this audio segment.".

### C.4.17 SINGING DETECTION

We aim for singing detection to go beyond simply identifying background music or relying on lyrics to determine whether singing is occurring. Instead, we seek to differentiate singing from normal speech by recognizing the distinct rhythm and melody of singing. To achieve this, we constructed our singing detection dataset using RAVDESS ( (Livingstone & Russo, 2018)), which consists entirely of a cappella performances where the context is unrelated to the singing. The task is then defined as: "Is there singing in this audio clip?Please answer by yes or no"

### C.4.18 COVID-19 RISK DETECTION

We use the Virufy COVID-19 Open Cough Dataset (Chaudhari et al., 2020) to construct our evaluation set. We classify the samples with positive test results as COVID-19 at risk, while those with negative results are classified as not at risk. And the task was set as: "Please listen to the following cough sound and determine whether the person is at risk of having a COVID-19 infection. Respond with 'yes' or 'no'"

### C.4.19 COUGH TYPE CLASSIFICATION

We use the COUGHVID (Orlandic et al., 2021) dataset to construct our evaluation set. We only utilize the data that has been assessed by experts, which falls into two categories: evaluations by four experts and evaluations by one expert. We prioritize samples where three out of four experts agree, and then we use samples rated as "good" by the single expert. In this task, we ask the model to distinguish whether the cough is a wet cough or a dry cough. And the task was set as: "Please help me determine whether the cough in this audio segment is a dry cough or a wet cough. Please respond with 'wet' or 'dry'."

### C.4.20 COUGH ORIGIN DIAGNOSIS

We use the COUGHVID (Orlandic et al., 2021) dataset to construct our evaluation set. We only utilize the data that has been assessed by experts, which falls into two categories: evaluations by four experts and evaluations by one expert. We prioritize samples where three out of four experts agree, and then we use samples rated as "good" by the single expert. In this task,The origins we tested include'COVID-19', 'healthy cough', 'lower infection', or 'upper infection'. And the task was set as: "Please help me determine the infection origin of the cough in the following audio segment. Choose from 'COVID-19', 'healthy cough', 'lower infection', or 'upper infection'."

### C.4.21 COUGH SEVERITY ASSESSMENT

We use the COUGHVID (Orlandic et al., 2021) dataset to construct our evaluation set. We only utilize the data that has been assessed by experts, which falls into two categories: evaluations by four experts and evaluations by one expert. We prioritize samples where three out of four experts

agree, and then we use samples rated as "good" by the single expert. In this task, the severity levels we tested include: 'pseudocough', 'mild', or 'severe'. And the task was set as: "Please help me assess the severity of the cough in the audio segment. Choose from 'pseudocough', 'mild', or 'severe'."

### C.4.22  SPOKEN ENGLISH COACH

We used speechocean762 (Zhang et al., 2021) to construct our evaluation set.In selecting our evaluation set, we aimed to include a wide variety of pronunciation errors by prioritizing sentences with poorer pronunciation quality. Here is how we built our sentence collection:

We started by selecting 207 sentences based on word stress errors (score == 5). Next, we chose 6 sentences with incomplete sentences or error-containing words (score < 10). Then, we selected 332 sentences with poor fluency (score <= 5). Following that, we picked 85 sentences with poor rhythm (score <= 5). Subsequently, we chose 179 sentences with low accuracy (score <= 5). Finally, we selected 40 sentences from each accuracy score level where the scores were higher. This process resulted in a final set of 1009 sentences. When constructing the ground truth for the answer output, we adopted the descriptions used in the original project for dataset scoring, and by concatenating these descriptions, we formed the final answer.

### C.4.23  VOICE DETECTIVE

When constructing the Voice Detective evaluation set, we used the SpeechAccentArchive dataset (Weinberger, 2013). The primary reason for choosing this dataset is the difficulty in obtaining a large amount of similar data, which significantly reduces the risk of data leakage. This constraint also compels researchers to focus more on factors such as the age and background of the users within the dataset.

### C.5  CREDIBILITY VERIFICATION

### C.5.1  ASR FOR LEGAL TERM

Since the legal vocabulary we selected, can be found in open-source code, is not complex, we introduced only one evaluator with a background in legal education, who is a native Mandarin speaker. The remaining three evaluators are regular native Mandarin speakers, making a total of four evaluators. If any one of the evaluators deems the speech quality insufficient, the corresponding speech will be discarded. The specific details of the evaluators are as follows:

Evaluator 1: 24 years old, male, graduated with a bachelor's degree from China University of Political Science and Law and is currently a master student at China University of Political Science and Law. Native Mandarin speaker.

Evaluator 2: 20 years old, female, currently an undergraduate student at Hubei University of Technology. Native Mandarin speaker.

Evaluator 3: 20 years old, female, currently an undergraduate student at Wuchang Shouyi University. Native Mandarin speaker.

Evaluator 4: 26 years old, male, high school graduate. Native Mandarin speaker.

### C.5.2  ASR FOR LEGAL MEDICAL

Due to the involvement of some medical terminology, this paper selected two evaluators with a medical background, along with two additional evaluators without a medical background. All of them are native Mandarin speakers. Similarly, if any one of the evaluators finds an abnormality in the speech, it will be discarded. The specific details of the evaluators are as follows:

Evaluator 1: 33 years old, female, graduated with a bachelor's degree from Hebei Medical University and has since been working in a medical-related field. Native Mandarin speaker.

Evaluator 2: 26 years old, female, completed an eight-year integrated program (continuously pursued both bachelor's and master's degrees) at Hebei Medical University and continues to work in a medical-related field. Native Mandarin speaker.

Evaluator 3: 25 years old, male, graduated with a bachelor's degree from Beijing Forestry University and is currently a graduate student at Beijing University of Posts and Telecommunications. Native Mandarin speaker.

Evaluator 4: 54 years old, male, graduated from a technical secondary school. Native Mandarin speaker.

### C.5.3 EMOTION TRANSLATION

We selected four evaluators and recorded their English proficiency. Similarly, if any one of the evaluators finds an abnormality in the speech, it will be discarded. The specific details of the evaluators are as follows:

Evaluator 1: 25 years old, female, graduated with a bachelor's degree from China Jiliang University and a master's degree from Beijing University of Posts and Telecommunications. English proficiency: CET-6.

Evaluator 2: 25 years old, female, graduated with both a bachelor's and a master's degree from Beijing University of Posts and Telecommunications. English proficiency: CET-6.

Evaluator 3: 23 years old, male, graduated with a bachelor's degree from Beijing Institute of Technology and is currently a PhD student at The Chinese University of Hong Kong, Shenzhen. English proficiency: IELTS Academic score: 6.5.

Evaluator 4: 28 years old, male, graduated with a bachelor's degree from Beijing University of Posts and Telecommunications and is a PhD student at Beijing University of Posts and Telecommunications. English proficiency: CET-6.

## D EXPERIMENT DETAILS

Below, we will divide the experiment details into four parts: details of human evaluation in Sec. D.1, details of model evaluation in Sec. D.3, and metric details in Sec. D.4.

### D.1 HUMANS EVALUATION DETAILS

In this section, we will introduce the participant information of our humans performance evaluation in Sec. D.1.1 and present the results of the consistency test for the result in Sec. D.1.2.

#### D.1.1 PARTICIPANT INFORMATION

Evaluator 1: Female, 28 years old, graduated with a bachelor's degree from East China Normal University, PhD from the Institute of Physics CAS. Evaluator 2: Female, 26 years old, graduated with a bachelor's degree from Beijing Normal University, master's degree from Shanghai Jiao Tong University. Evaluator 3: Male, 29 years old, graduated with a bachelor's degree from Beijing University of Chemical Technology, PhD from Beijing University of Posts and Telecommunications. Evaluator 4: Male, 27 years old, graduated with a bachelor's degree from Xidian University, currently pursuing a PhD at Singapore University of Technology and Design (SUTD).

#### D.1.2 CONSISTENCY TEST

To verify the consistency of the humans evaluation, We focus on objective multiple-choice questions. we calculated the proportion of questions where all three volunteers selected the same option, as well as the proportion where all four volunteers chose the same option, relative to the total number of questions. These proportions are shown in Tab. 13.

It is also important to note that, since our testers are only proficient in English, they were unable to complete the Language Identification task.

Table 13: Consistency for Humans Evaluation

| Task | Accuracy | Num of Questions | Proportion (3 Evaluators Same) | Proportion (4 Evaluators Same) |
|------|----------|------------------|--------------------------------|--------------------------------|
| Volume Perception | 100.00% | 40 | 100.00% | 100.00% |
| Pitch Perception | 96.25% | 40 | 100.00% | 95.00% |
| Binaural Effect Perception | 100.00% | 40 | 100.00% | 100.00% |
| Ambient Noise Detection | 91.88% | 40 | 100.00% | 87.50% |
| Acoustic Scenes Classification | 90.28% | 180 | 97.22% | 93.89% |
| Speaker's Age Prediction | 52.59% | 60 | 76.67% | 46.67% |
| Speaker's Gender Recognition | 97.50% | 40 | 100.00% | 100.00% |
| Speech Emotion Recognition | 50.71% | 140 | 94.29% | 85.71% |
| Cappella Emotion Recognition | 62.25% | 100 | 92.00% | 68.00% |
| Emotion Intensity Perception | 97.50% | 40 | 100.00% | 95.00% |
| Singing Detection | 98.13% | 40 | 100.00% | 97.50% |
| COVID-19 Risk Detection | 60.63% | 40 | 70.00% | 17.50% |
| Cough Type Classification | 52.50% | 40 | 77.50% | 22.50% |
| Cough Origin Diagnosis | 32.19% | 80 | 28.75% | 2.50% |
| Cough Severity Assessment | 45.42% | 60 | 45.00% | 11.67% |

### D.1.3 DEFICIENCY IN HUMANS EVALUATION.

During the Humans Evaluation process, we were unable to find a native English speaker, but all participants involved in the evaluation are proficient English users. We also could not find individuals who are proficient in multiple languages, which made it difficult to conduct a Humans Evaluation for the Language Identification task.

### D.2 GPT-4O MANUAL TEST DETAILS

To evaluate GPT-4o advanced speech mode, we synthesized the text instructions from each level test into audio as instructions. We sampled 80 samples for each task. Each test is played by a person using a speaker to GPT-4o running on an iPhone 15 device. Since the GPT-4o advanced speech mode supports speech-to-speech conversion, we manually process its text output for evaluation. We find that GPT-4o currently tends to refuse to answer some audio tasks, we treat them as being unable to follow instructions.

### D.3 MODELS EVALUATION DETAILS

We divide our experimental details into two sections: the model replication platform in Sec. D.3.1, and the model replication details in Sec. D.3.2.

### D.3.1 EXPERIMENTAL PLATFORM

In this paper's experiments, all servers used are equipped with an Intel® Xeon® Platinum 8358 CPU @ 2.60GHz as the core processor. Each server is loaded with eight NVIDIA A800-SXM4-80GB graphics cards, and each model runs with exclusive use of one A800 card.

### D.3.2 MODELS REPLICATION DETAILS

In this paper, we aim to select the 7B-level versions of various models wherever possible. However, due to the differences between various models, it is difficult to ensure that their parameter counts are exactly the same.

**Qwen-Audio** For the Qwen-Audio model (Chu et al., 2023), we reproduced the model using its open-source code.

**Mu-LLaMA** In the process of implementing the model Mu-LLaMA (Liu et al., 2024b) , this paper used the LLama2-7B-chat (Touvron et al., 2023) checkpoint to maintain consistency with the original paper, and utilized the open-source MU-LLaMA checkpoint provided.

**GAMA** Since the primary focus of this paper is to test the audio understanding capabilities of the GAMA model (Ghosh et al., 2024b), we consulted with the authors and selected the 'state4epoch2' checkpoint over the 'state5epoch2' checkpoint, as it has superior audio comprehension abilities

**SALMONN** For the SALMONN model (Tang et al., 2023), we tested the model using its open-source code.

**Qwen2-Audio** For the Qwen2-Audio model (Chu et al., 2024), we reproduced the model using the 7B version of its open-source code.

### D.4 Matrix

We have designed three metrics: WER, the accuracy for objective multiple-choice questions, and GPT-4o scoring, specifically targeting ASR tasks, objective multiple-choice questions, and subjective responses. This section will provide detailed explanations. For an overview, please refer to the following Tab. 14.

Table 14: Metrics for Each task

| Task | Metric |
|---|---|
| Language Identification | 5-Categories Acc |
| Speech ASR | WER |
| Song ASR | WER |
| Volume Perception | 2-Categories Acc |
| Binaural Effect Perception | 2-Categories Acc |
| Ambient Noise Detection | 2-Categories Acc |
| Speaker's Age | 3-Categories Acc |
| Speaker's Gender | 2-Categories Acc |
| Sound Event Classification | 9-Categories Acc |
| Singing Detection | 2-Categories Acc |
| Speech Emotion Recognition | 7-Categories Acc |
| Song Emotion Recognition | 5-Categories Acc |
| Emotion Intensity Perception | 2-Categories Acc |
| Disorder Detection | 2-Categories Acc |
| Speech Disorders Detection | 2-Categories ACC |
| COVID-19 Risk Detection | 2-Categories ACC |
| ALS Detection | 2-Categories ACC |
| Accent Detection | 11-Categories Acc |
| Emotion Translation | GPT Score |
| Spoken English Coach | GPT Score |
| Voice Detective | GPT Score |

#### D.4.1 WER for ASR

The Word Error Rate (WER), a key metric for gauging the effectiveness of Automatic Speech Recognition (ASR) systems, quantifies the divergence between an ASR system's output and a reference transcript. It assesses the total error rate by tallying the number of insertion, deletion, and substitution operations needed to align the ASR output with the true reference text.

While computing the WER, certain variances in word usage, like "I am" compared to "I'm," may be seen as semantically equivalent by human standards but are flagged as errors by computational algorithms. Thus, a standardization process is essential prior to WER calculation to make both texts directly comparable. The methodology for this standardization, akin to what is employed in the Whisper (Radford et al., 2023) framework, has been detailed in a related research paper. It has been demonstrated that this approach exerts negligible influence on the assessment of WER outcomes when tested against the LibriSpeech (Panayotov et al., 2015) dataset, which was utilized in our paper.

For cases where the error rate exceeds 100% (i.e., WER is over 1), we mark them in our experimental records as having significant recognition errors. Such data will not be included in the calculation

of the final average WER. In the final record of the experiment, we will focus on two key metrics: first, the ASR completion rate, which is the percentage of data with a WER less than 1; second, the mean WER of the completed portion, which is the average WER of data with a WER less than 1. If the mean WER of the completed portion does not decrease to below 0.8, we will conclude that the model lacks effective automatic speech recognition (ASR) capabilities and document this finding in detail in the experimental results.

The implementation details regarding WER (Word Error Rate) can be found in our publicly available code.

### D.4.2 ACCURACY FOR OBJECTIVE MULTIPLE-CHOICE QUESTIONS

A selection is considered correct only if the model chooses the correct answer and no other options. If the model selects two or more options, even if the correct one is included, it will be deemed incorrect.

### D.4.3 ACCURACY FOR ASR ON TERMS

Since in these tasks we primarily assess the ability of speech LLMs to transcribe terms, we consider a response correct as long as the correct term is included in the speech transcription, without focusing on the accuracy of other parts of the sentence.

### D.4.4 SCORING FOR SUBJECTIVE RESPONSE QUESTIONS

In our experiments, we used GPT-4o to assist in evaluating the results. The specific prompt used is as follows.

**Prompt for Emotion Translation**

I currently need your assistance in evaluating some translations. The most suitable translations should incorporate the corresponding emotions appropriately. The scoring ranges from 0 to 4. I will provide you with the original English sentence, the associated emotional label, and the suggested translation, allowing you to score them based on the context.

Here are some examples:

[Here are some scoring examples. Due to space limitations, we have omitted them in this section. You can find the details in the code we have made available.]

Now Answer:[ANSWER]

Label:The original sentence is: <emotion>[SENTENCE] The suggested translation is: [SUGGESTION].

Please provide your score.

**Prompt for Spoken English Coach**

I now need you to help me evaluate some Answers for accuracy. You need to evaluate and score in the order of overall pronunciation, fluency, prosody, words that are mispronounced, and words that have incorrect stress. The score ranges from 0 to 4. Here are the specific scoring rules: You need to first check if the evaluation of overall pronunciation in the Answer matches the Label. If they do not match, give a score of 0 and continue with the evaluation; if there is no relevant description, also give a score of 0 and continue with the evaluation; if it is correct, add 1 point and continue with the evaluation.

For fluency and prosody in the Answer compared to the Label, award up to 1 point for each if completely correct, a partial score for partially correct, and no points if there is no relevant expression. Finally, check the descriptions in the Answer and Label regarding words that are mispronounced and words that have incorrect stress. Award 1 point only if all are correct. If part of the descriptions are correct, you can give a partial score, such as 0.33 points for one out of three correct descriptions. Here are some examples:

[Here are some scoring examples. Due to space limitations, we have omitted them in this section. You can find the details in the code we have made available.]

Now Answer:[ANSWER]

Label:[LABEL]

Please provide your score.

### D.4.5 PROMPT FOR VOICE DETECTIVE

I now need you to help me evaluate some Answers for accuracy. You should focus on whether the information about gender, place of birth, age, and native language in the Answer matches the Label, and provide a final rating. Award 1 point for each correct piece of information, with no points for incorrect information. Please give your score on a scale of 0 to 4. Here are some examples:

[Here are some scoring examples. Due to space limitations, we have omitted them in this section. You can find the details in the code we have made available.]

Now Answer:[ANSWER]

Label:[LABEL]

Please provide your score.

## E INSTRUCTION FOLLOW EXPERIMENT

### E.1 SPEAKER'S AGE PREDICTION

The instructions used in the experiment are as follows:

- **Instruction variation I** In which age group do you think the speaker's voice belongs?
- **Instruction variation II** What age category do you believe the speaker's voice fits into best?
- **Instruction variation III** Which age bracket do you feel corresponds to the speaker's voice?
- **Instruction variation IV** How old do you think the speaker sounds, based on their voice?
- **Instruction variation V** Which age range would you assign to the speaker's voice?
- **Instruction variation VI** What age range do you associate with the speaker's voice?
- **Instruction variation VII** Which age group do you think best describes the speaker's vocal characteristics?
- **Instruction variation VIII** What do you believe is the age range of the speaker judging by their voice?

The experimental results are recorded in Tab. 15.

Table 15: The impact of different prompts on age detection

| Prompt | Qwen-Audio | Qwen2-Audio | MuLLama | GAMA |
|---|---|---|---|---|
| Our benchmark instruction | 29.29% | 38.55% | 33.60% | 0.2% |
| Instruction variation I | 23.03% | 36.36% | 35.45% | 0.4% |
| Instruction variation II | 31.82% | 36.97% | 35.45% | 4.85% |
| Instruction variation III | 12.83% | 38.38% | 34.75% | 0.0% |
| Instruction variation IV | 4.44% | 43.03% | 31.31% | 0.2% |
| Instruction variation V | 28.89% | 37.37% | 33.03% | 0.1% |
| Instruction variation VI | 19.90% | 37.27% | 34.14% | 0.0% |
| Instruction variation VII | 6.57% | 36.77% | 30.81% | 0.3% |
| Instruction variation VIII | 26.77% | 41.11% | 28.67% | 0.4% |

### E.2 ACOUSTIC SCENES CLASSIFICATION

- **Instruction variation I** How would you detect the background sound in this audio clip?

- **Instruction variation II** What kind of ambient noise can be heard in this segment?
- **Instruction variation III** Can you describe the environmental sounds present in this audio?
- **Instruction variation IV** What background audio elements are featured in this segment?
- **Instruction variation V** What atmosphere is created by the sounds in this audio segment?
- **Instruction variation VI** Can you identify the ambient sound in this clip?
- **Instruction variation VII** What noises are occurring in the background of this audio?
- **Instruction variation VIII** What type of surrounding sound is present in this recording?

The experimental results are recorded in Tab. 16.

Table 16: The impact of different prompts on acoustic scenes classification

| Prompt | Qwen-Audio | Qwen2-Audio | MuLLama | GAMA |
|---|---|---|---|---|
| Our benchmark instruction | 18.84% | 27.67% | 5.07% | 12.05% |
| Instruction variation I | 13.05% | 35.68% | 1.91% | 0.00% |
| Instruction variation II | 8.97% | 13.73% | 5.91% | 0.36% |
| Instruction variation III | 4.29% | 9.66% | 0.00% | 0.94% |
| Instruction variation IV | 5.43% | 9.95% | 0.00% | 1.87% |
| Instruction variation V | 13.95% | 28.29% | 1.87% | 0.54% |
| Instruction variation VI | 15.32% | 21.87% | 2.02% | 0.25% |
| Instruction variation VII | 5.37% | 5.23% | 1.8% | 0.00% |
| Instruction variation VIII | 9.62% | 18.92% | 6.31% | 4.32% |

