# OpenReview forum: "Roadmap towards Superhuman Speech Understanding using Large Language Models"
_ICLR.cc/2025/Conference — ICLR 2025 Conference Withdrawn Submission_

### Official Review · Reviewer_Ps9Q · 2024-11-02

**Soundness:** 2
**Presentation:** 2
**Contribution:** 2
**Rating:** 3
**Confidence:** 4

**Summary:**

This paper curates a new benchmark that covers a range of speech tasks across difficulty levels (starting from basic recognition to advanced voice-assistant-like) and compares a set of speech LLMs on these tasks. It also provides some insights into why speech LLMs do poorly on certain types of tasks.

**Strengths:**

1. The paper provides a useful comparison between different speech LLMs on a host of tasks spanning difficulty levels.
2. It provides one of the first benchmark evaluations for GPT-4o.
3. Sections 4.2-4.4 are informative; these are common issues with speech LLMs but haven’t been properly reported on before, so it was good to see this discussion.

**Weaknesses:**

1. The paper suffers overall from overclaims that use the terms ‘superhuman’, ‘AGI’, and ‘surpassing human performance’ too loosely without properly defining them or providing experimental results to support. At its core, this paper is a useful speech LLM benchmark that analyzes speech LLM performance on increasingly difficult tasks (from level 1 to level 5 difficulty). To avoid misconceptions, I recommend the authors reduce the hype-like formulation using loaded terms like AGI and focus on making scientific claims. Takeaway 4 (lines 340-345) is an overclaim as well.
2. The task categorization is not very novel e.g. an existing benchmark like Dynamic SUPERB already categorizes by levels 1-3, which are called content, paralinguistics, and semantics respectively in Dynamic-SUPERB; this is acknowledged in Appendix A of this paper as well. The level 4 and 5 categories (speech specialist and ‘AGI’) are new, but this benchmark does not add enough tasks in those categories (only 4 COVID-based tasks in L4 and 2 tasks in L5), so these new levels are not covered sufficiently. The authors say they will be adding more tasks, but they are not included in this submission.
3. A class of baselines that I think are missing are text-based baselines that try to infer emotion/stress from the transcript of the utterance, and/or with additional automatically-extracted metadata from the speech. These baselines would help understand how much of the benchmark is solvable with text alone and how much requires speech data.
4. The writing for the benchmark description repeats the definition of each level in different places, losing out on space. I recommend making Section 2 much tighter so that some important experimental details (currently in appendices) can be included in the main text.

**Questions:**

1. Based on your findings from this benchmark, how do you think speech models could be improved? It would be helpful to add some action items for the speech community, in addition to the weaknesses you have found.

---

### Official Review · Reviewer_smAV · 2024-11-03

**Soundness:** 3
**Presentation:** 3
**Contribution:** 2
**Rating:** 5
**Confidence:** 4

**Summary:**

In this work, the authors propose an intuitive roadmap to guide the development of speech large language models (LLMs) towards superhuman understanding capabilities. These levels range from basic transcription to advanced understanding of complex acoustic knowledge. To validate this roadmap, the authors introduce the **SAGI Benchmark** to evaluate model performance across tasks representative of each level. The evaluation of both speech LLMs and human performance on this benchmark reveals insights into current capabilities and limitations of existing models, which can guide future model development.

**Strengths:**

1. The proposed roadmap is an innovative approach to organizing the progression of speech LLMs, characterizing speech understanding in these models to five distinct levels.

2. The SAGI benchmark is proposed to align with the roadmap, and the evaluation of speech language models on this benchmark highlights a variety of limitations of these models, covering the incapability of instruction following and acquisition of acoustic information.

**Weaknesses:**

1. The current version of the SAGI benchmark is limited in its coverage, particularly for Levels 4 and 5, which involve advanced and generalized speech comprehension tasks.

2. The tasks proposed in the SAGI benchmark do not align well with the definitions for each level as outlined by authors. Specifically, the current level 5 tasks mainly focus on pronunciation scoring and native language identification, whereas the criteria for level 5 (lines 265-266) emphasize encouraging creativity and diverse thinking. I believe these tasks do not fully satisfy the intent.

3. I am concerned about the effectiveness of both the human baseline and GPT-4o evaluation. For the human baseline, I find it problematic that all the speech language models surpass humans on the “auto-speech recognition” task at level 1, which seems highly unrealistic and raises questions about the validity of this baseline. Additionally, the GPT-4o evaluation was conducted manually without using an API, making it especially sensitive to environmental sounds and the recording quality of the devices used, which casts doubt on the reproducibility of this baseline.

**Questions:**

In this section, I raise several questions and comments regarding the SAGI benchmark, evaluation results, and paper presentation.

**SAGI Benchmark**
1. The paper mentions that the benchmark tasks are not yet fully complete. Could you clarify which additional tasks will be included in the final version?
2. While the paper primarily addresses speech understanding, I find that music-related tasks in the benchmark are somewhat limited. Adding tasks like vocal technique detection could enhance the diversity of the benchmark.
3. As I noted in the weaknesses section, some current tasks assigned to Level 5 don’t fully meet the criteria for that level. I suggest that the authors provide more examples of tasks that clearly qualify as Level 5.

**Evaluation Results**
1. In Table 3, there’s a row labeled "Hallucination Rate" under the task “auto-lyrics transcription.” However, there is no further explanation of this metric in the main text or in the table notes. Could you clarify what this row indicates?

2. Figure 4 includes two legends, "Different" and "Repeated." While the paper explains that "Different" means the spoken content is the same but the speaker’s characteristics (such as gender or emotion) differ, it lacks an explanation for "Repeated." Could you provide details on the experimental setup for this legend?

**Paper Presentation**
1. I suggest positioning the “*” mark next to the task names in the "Task" column instead of next to each model's performance. This adjustment would enhance the table’s clarity and better align with other notations.

2. Table 4 could be misleading in terms of which model performs better, as it doesn’t provide the metrics for these tasks. It might be helpful to bold the best results in each row to make the comparisons easier to follow.

---

### Official Review · Reviewer_Hobr · 2024-11-04

**Soundness:** 3
**Presentation:** 2
**Contribution:** 2
**Rating:** 3
**Confidence:** 4

**Summary:**

This paper outlines a roadmap for achieving superhuman speech understanding by defining five levels of speech and audio understanding tasks and establishing a benchmark based on this framework. The first level involves automatic speech recognition, a task that has been successfully completed by various multimodal large language models (LLMs). As we move up to higher levels, the tasks require more contextual information and knowledge. In addition to mapping out these levels, the paper presents preliminary benchmarks for each level, comparing existing models with human performance. This information is invaluable and includes detailed analyses highlighting how far we are from reaching the established baseline.

**Strengths:**

- High-level (superhuman) speech understanding based on LLMs is a core topic for realizing conversational AI. However, we don't have an established roadmap and benchmark toward this goal, and this attempt is very valuable.
- The benchmark has human performance, and we can evaluate our models with its superhuman degree in each task

**Weaknesses:**

- The paper does not address speech generation tasks (e.g., Text-to-Speech (TTS) or dialogue system responses) and lacks completeness as a roadmap or benchmark for the field.
- Certain terminology choices, such as "paralinguistics," may not align with conventional use within the speech community, leading to potential misinterpretation. For instance, "paralinguistics" typically encompasses emotional aspects of speech, as illustrated by the Computational Paralinguistics Challenge organized by ISCA Interspeech, which includes emotion recognition tasks. This divergence in terminology creates ambiguity and suggests the roadmap may lack refinement in this respect.
  - Due to these terminology issues, distinctions between levels—specifically L2 and L3 tasks—are unclear. Some tasks categorized at L2, such as pitch perception, seem more complex than certain L3 tasks, such as gender recognition.
- I am skeptical of the overall categorization scheme, as some tasks listed (e.g., acoustic scene classification and L4 tasks) are not strictly "speech" tasks but are more accurately described as audio processing tasks. Additionally, L4 tasks are likely to lack significant speech semantics, making them atypical for speech understanding. However, if L4 included tasks involving doctor-patient conversations with consultative dialogues, it could be more suitable for speech understanding.
- The paper overlooks relevant existing speech and audio benchmarks, such as the SUPERB and HEAR benchmarks, which could strengthen the study’s scope and positioning.
- The benchmark section itself feels incomplete. As noted in the footnote, tasks for Levels 4 and 5 have not been fully identified or outlined.
- Finally, the analysis sections appear somewhat disconnected from the roadmap (especially Sections 4.3 and 4.4), with limited cohesion between the analysis content and the proposed roadmap.

**Questions:**

- Page 2, Itemization of Levels: The three levels listed here conflict with the five levels in the roadmap, which may create confusion. Given the noted ambiguity between Levels 2 and 3, I recommend refining the levels and corresponding tasks for clearer distinction.
- Page 4, Level 2 Terminology: Consider using terminology more specific to physical or acoustic features rather than "paralinguistics," as this section pertains more to physical or acoustic information than traditional paralinguistic elements.
- Page 4, Level 3 - Citation Needed: The statement, “Interestingly, even some higher animals, like pet dogs, can perceive these types of non-semantic information,” would benefit from supporting citations to strengthen the claim.
- Page 5, Lines 218-219 - Citation for Gladwell Reference: Please add a citation for “The 10,000-Hour Rule” from Malcolm Gladwell’s Outliers.
- Table 2 - License Information: Including license details in Table 2 would be helpful, as this is essential for widely-used benchmarks.
- Table 3 - Indicate Superiority with Symbols: Adding arrows (↑, ↓) in Table 3 to indicate the direction of performance metrics (accuracy vs. error) would make it easier to distinguish superior and inferior results.
- Page 6, Section 3.2 - Clarify “Manually” in Testing GPT-4o: The term “manually” in this context is vague. Please clarify the process of how GPT-4o was used in the advanced speech mode.
- Experimental Findings - Limited Novelty: The experimental findings seem somewhat elementary. For instance, GAMA’s superior performance in the audio task and speech LLMs’ strengths in term recognition (Section 4.1) are unsurprising.
- Tables 3 and 4 - Consistency of Units: There are inconsistencies in the representation of units (e.g., omission of % in some tasks). If this is intentional, please clarify the rationale.

---

### Official Review · Reviewer_M7Vk · 2024-11-05

**Soundness:** 1
**Presentation:** 4
**Contribution:** 2
**Rating:** 3
**Confidence:** 4

**Summary:**

The paper presents a vision for benchmarking speech language modeling within speech understanding tasks, proposing a roadmap in five stages with an associated benchmark. In comparison to previous benchmarks, the SAGI benchmark uniquely addresses paralinguistic cues and abstract acoustic knowledge. The paper extensively discusses five models—Qwen2-audio, Mu-llama, GAMA, SALMONN, and GPT-4o—accompanied by in-depth analysis.

**Strengths:**

- The paper contributes significantly by framing speech language modeling tasks within a structured roadmap and providing a comprehensive analysis of the five-stage formulation.
- Experiments were conducted to verify the progressive nature of the roadmap, revealing current limitations and potential directions for the future development of speech language modeling. The insights gained generally align with the proposed roadmap vision, yielding valuable takeaways.

**Weaknesses:**

The primary concern with the paper is its over-claims and the lack of a well-defined conceptual foundation in its design. While the authors’ efforts to propose an abstraction for speech language modeling in the LLM era are commendable, the current formulation—particularly the high-level abstractions—lacks convincing theoretical grounding. Specific issues include:

- In Section 2.2, the paper outlines the philosophy of the proposed roadmap; however, the definitions of each level lack theoretical depth, serving more as categorizations than rigorously justified stages. Even if the defined levels are accepted, the sequence of the first three levels is debatable. For instance, Level 2, which addresses basic paralinguistic perception, could logically precede speech recognition, as suggested by previous studies. If the framework is to be positioned as a roadmap, further justification is essential to clarify the rationale behind the level definitions and their order with enough theoretical background support.

- The benchmark relies on a limited selection of open-source corpora, but the chosen tasks lack alignment with the intended focus. For example, at Level 4, the authors utilize the COVID-cough dataset, which is not typically associated with speech tasks (usually classified as general audio tasks given its limited language-related content). While the authors aim to exclude general audio/music-related tasks, the current framework inadvertently includes such elements, particularly when analyzing music or environmental sounds. More robust justification is needed to delineate the scope of the proposed method. At Level 5, computer-assisted language learning traditionally viewed as analogous to ASR, is positioned at an AGI level, which may be overly ambitious. As the paper discusses, speech AGI aims to surpass human performance in speech-based tasks, but the selected tasks fall short of representing this objective effectively.

**Questions:**

Beyond the main concerns outlined above, the reviewer has the following questions and comments:

- Language plays a crucial role in speech modeling. The current tasks predominantly involve English, with a few tasks in Chinese, which complicates the differentiation between language-specific effects and model reasoning. It may benefit the authors to consider multilingual approaches within the modeling scope or by limiting the scope and focusing on a single language.
- The validity of using human performance as a benchmark in some tasks is questionable. Providing statistical significance for human evaluations would enhance credibility, especially given the limited number of questions and participants. Furthermore, involving students without relevant background knowledge (as inferred from participant details) in cough analysis raises concerns about the reliability of the human benchmark.

**Details Of Ethics Concerns:**

The benchmark is a collection of existing corpora, where detailed license information is not included.

---

### Note · Authors · 2024-12-16

I have read and agree with the venue's withdrawal policy on behalf of myself and my co-authors.